# Perception of Realism and Acquisition of Clinical Skills in Simulated Pediatric Dentistry Scenarios

**DOI:** 10.3390/ijerph191811387

**Published:** 2022-09-09

**Authors:** Begoña Bartolomé Villar, Irene Real Benlloch, Ana De la Hoz Calvo, Gleyvis Coro-Montanet

**Affiliations:** Department of Preclinical Dentistry, Faculty of Biomedical Sciences, European University of Madrid, 28670 Madrid, Spain

**Keywords:** high fidelity, simulation training, clinical competencies, manikin

## Abstract

Simulation, depending on the modality and fidelity of the scenarios, is an important resource for clinical teaching and achievement of learning outcomes in dentistry. The objectives of this study were to compare the degree of realism perceived by students and teachers in a simulated scenario, and to assess the level of competence acquired by the students. Method: In the Pediatric Dentistry course, eight clinical scenarios were carried out, each one using a modified Erler Zimmer child simulator (handmade), a professional actress and two students (dentist and assistant) on the same pediatric dentistry case consisting of a pulp abscess in tooth 8.5. A total of 114 students in the 4th year of dentistry studies participated in the pediatric dentistry course. Questionnaires with Likert-type answers were elaborated to evaluate the educational intervention, applying them before and after the simulation. Results: The realism best valued by the students was that of the simulated participant and the worst that of the manikin, the latter being strongly related to the realism of the office. It was observed that students’ perception of clinical competence increased as the overall realism of the scenario increased (*p*-value = 0.00576). Conclusion: This research suggests that the creation of scenarios using handmade mannequins and simulated participants achieves a high level of realism, increasing the level of clinical competence perceived by dental students.

## 1. Introduction

The acquisition of motor skills is not enough for an adequate and complete dental training. Certain non-technical skills should be added to that training in order to achieve certain clinical skills. Some of these non-technical skills may be critical thinking, teamwork, decision making, leadership, and situational knowledge [1]. For this reason, simulation is perhaps, at present, one of the best pedagogical methods to achieve the integration of both motor and non-technical skills in a safe environment. Through simulation, the student is able to recreate repeatable real situations and make mistakes without causing harm to the patient. It offers the possibility of training before moving on to the clinic [2], combining theoretical knowledge, motor skills and developing competencies. This way of training leads to high satisfaction, both for the student and the teacher, when the proposed learning objectives are achieved [1]. 

These learning objectives will determine both the modality and the level of fidelity of the simulation that will be used. The modality of the simulation refers to the methodology used and the equipment necessary for it. This way it is possible to distinguish “tasks trainers” and high/low fidelity scenarios. The level of fidelity will depend on the use of mannequins with more or less sophisticated software and the use of standardized or virtual patients [3]. With new technologies, it has been possible to create scenarios using digital methods, achieving greater immersion of the students in the scenario [4]. 

There is controversy about which modality best facilitates learning. It is generally believed that low fidelity builds knowledge, medium fidelity facilitates the acquisition of competencies and high fidelity generates the action [3]. For some authors, low-fidelity models obtain the same results as high-fidelity models [5]; for others, high-fidelity models are preferable, especially for more advanced learners [6], while some consider medium fidelity to be sufficient for the acquisition of integrated skills [1,7]. 

Fidelity refers to the degree of realism developed in the scenario, accurately simulating the real situation, which helps to increase the student’s immersion. It is necessary to differentiate between physical, semantic or conceptual and phenomenological or psychological fidelity [8]. Some authors even add a functional scale (mainly associated with psychomotor tasks) and a sociological scale (referring to interprofessional interactions) [3]. Physical fidelity is based fundamentally on the veracity of the simulator, semantic fidelity refers to the conceptual field between what happens and the consequences that follow, and phenomenological (psychological) fidelity describes an emotional aspect, i.e. whether the situation is credible [8]. Other authors distinguish the so-called physical fidelity of the environment, in relation to the context in which the simulator is situated, and the psychological fidelity which refers to the extent to which the situation is perceived as real [9]. The problem is that the different types of fidelity may interfere with each other and may be complementary or cancel each other out [3]. Thus, the category of fidelity that we are most interested in emphasizing with each simulation must be adapted according to the objectives to be achieved on that specific case.

Another factor to consider when choosing the most appropriate modality and degree of fidelity—apart from the learning objectives—is the economic cost of the simulation. Due to the high cost in time and resources generated by the creation of cases for simulation, emphasis should be placed on the development of human aspects, which are the most difficult to acquire through regular practice [10]. In order to reduce costs, some authors propose that resident students may act as teachers in the instruction, obtaining equally satisfactory results [11,12]. Another option is the creation of scenarios among classmates instead of using standardized patients [5]. This also generates a decrease in stress among students and increases excitement and enthusiasm by making them direct participants in the learning process [13]. On the other hand, there are studies that prefer the use of standardized patients versus mannequins because they confer greater realism [14] and lead to significant differences in performance and in the acquisition of most skills [15].

We can therefore say that the choice of the best modality of simulation and its degree of fidelity should be based fundamentally on the perception of the professionals who design the scenario [16]. However, the availability of equipment, the objectives and desired results, the level of preparation and previous knowledge of the students, their learning styles and motivation [3], as well as the cost-effectiveness of their inclusion in the programs should be considered [17].

The objectives of this study were to compare the degree of realism perceived by students and teachers in a simulated scenario and to assess the level of competence acquired by the students.

## 2. Materials and Methods

An analytical and comparative observational study was carried out analyzing the levels of realism perceived by students and teachers in simulated pediatric dentistry scenarios with a handmade simulator of a 6-year-old child, linking the overall results of these perceptions of realism with the perception of clinical competence of the sample of participating students. 

The simulator was tested in a previous simulated scenario and its level of realism was measured by 2 expert teachers with the validated tool ProRealSim v.1.0, obtaining a realism index of 5.1 (medium) for the manikin and considering it suitable for teaching application.

The simulated case consisted of a girl who came to the dentist’s office with pain and pulp abscess at the level of the lower right second primary molar. The scene was carried out in the simulated dental office with a professional actress in the role of the mother, the handmade manikin in the role of a 6-year-old girl and two students in the roles of the dentist and assistant. The learning objectives of the scenario were: To teach the techniques for managing the behavior of the pediatric patient and to elaborate a correct diagnosis and treatment of the case.

### 2.1. Student Sample

The educational intervention was applied to a population of 283 4th year dental students in the subject of pediatric dentistry. These students were distributed in 8 scenarios in which the same environment, professional actress and manikin were used. The students’ ratings were measured using a Likert scale with values from 1 to 10 to obtain estimates with an accuracy of 95% and an error of no more than 0.25 points. To obtain this value, a pilot sample was previously taken from which the standard deviation was estimated. After using the known formula for the minimum sample size: n > (1.96·σ)^2^/e

It was decided to take a sample of 114 students who assessed their perception of clinical competence after the scenario and the levels—partial and global—of realism perceived during the simulated scenario. After the pilot sample study—calculated by means of a formula—the sample of 114 students was considered significant. Four variables of realism were studied: scenographic (realism of the dental office), simulator (realism of the manikin), simulated participant (realism of the actress in the role of the mother) and global realism). Table 1 shows the questionnaire applied.

### 2.2. Teacher Sample

The realism of the scenarios was determined by using the validated ProRealSim v.1.0 tool (DESMONDO S.L. Madrid, Spain). To reduce the subjectivity of a single evaluator, two realism evaluators with a high level of experience in the design of similar activities were used. It was considered a sufficient and reliable sample to carry out the evaluation for several reasons: their level of in-depth knowledge of the subject as expert designers, that they had a consolidated culture of realism measurement and that both were trained in the use of the same ProRealSim v.1.0 tool, validated by statistical analysis (correlations, Cronbach’s Alpha and Guttman’s Lambda 6 index).

The measurements provided by both evaluators were averaged to obtain a final realism index. 

Thus, and according to the principles of validity and reliability in educational evaluation, with the support of mathematical indexes, and using the gold standard of a statistical index measured with expert criteria (the teachers’ evaluation), comparisons were made with the students’ perception for the scientific analysis of the subject.

### 2.3. Statistical Methods

Descriptive summaries were made using numerical summaries for numerical variables, and frequency tables for factors. Differences between the different groups studied were contrasted using the Kruskal–Wallis test (due to the lack of normality). 

For differences between the variables studied, we used the *t*-test for paired samples, the Analysis of Variance (ANOVA) test for repeated means (under the hypothesis of normality) or the Friedman test (when the hypothesis of normality was not fulfilled). Normality was tested using the Shapiro–Wilk test. 

Prediction models for a numerical variable were developed using linear regression models. The F test and the coefficient of determination R^2^ were used to measure the significance of these models. The relationship between numerical variables was determined using Spearman’s correlation coefficient and the correlation test. Correlation and box plots were used for a better visualization of the results. Results were considered significant for *p*-values less than 0.05. All analyses were carried out with R statistical software (version 4.1.1) (Statistical Service of the Universidad Autónoma de Barcelona, Bellaterra, Spain) using the RStudio environment. Graphical representations were made with the ggplot package and associated packages.

Eight scenarios on the same case with the same dimensions of realism (professional actress in the role of mother, handmade simulator and Gessel camera) were analyzed in a simulation center with more than 5 years of experience in the application of a clinical simulation program. Thus, the same scenario was repeated 8 times by different groups of students to obtain perception records with the greatest variability possible.

Since it was the same case represented repeatedly 8 times, the analysis of the data obtained was carried out together and not by scenarios, as corresponds to studies of this type where the casuistic variation is null or insignificant. The combined analysis is justified by the fact that each scenario contemplates the same conditions that qualify it as the same repeated experience.

## 3. Results

### 3.1. Student Evaluation

Figure 1 shows the absolute frequencies corresponding to the students participating in each of the scenarios, represented by a bar graph, with the highest number of individuals in scenario 3.

Next, the 4 variables that measure the realism perceived by the student (scenographic, simulator, simulated participant and global realism) are analyzed according to the perception of the participating students, both in global terms and for each of the scenarios (Table 2).

According to the ANOVA test for repeated measures (which yielded significant differences, *p*-value = 1.24 × 10^−^^2^^3^), the mean ratings of realism differed for the 4 categories analyzed. The significantly highest average rating was for perceived realism of the simulated participant (8.99), followed by overall realism (8.15), office realism (7.83) with high significance and, finally, perceived realism of the manikin (6.79), which outperformed the initial testing of the simulator by the teachers. These results are represented graphically in the box plot in Figure 2. In particular, it can be seen that the box relating to the realism of the manikin is clearly lower than the rest of the boxes.

As for the Kruskal–Wallis test, significant differences were found for all variables, except for the perceived realism of the simulated participant.

### 3.2. Teacher Evaluation

Next, the data on realism achieved, as measured by the teachers, are analyzed globally. Table 3 shows the mean values of each perception, together with the standard deviation and the result of the Friedman test, which contrasts whether the mean ratings coincide for the four variables. As a result, the average rating is not the same for the four ratings performed. The average ratings are higher for the perceived realism of the participant, which is clearly higher than for the overall perceived realism, perceived realism of the office and of the manikin. The graphical representation of these four variables is shown in Figure 3. It can be clearly seen that the realism of the manikin is the lowest, and the realism of the participant is the highest.

### 3.3. Comparison between Students and Teachers

The following is a comparative study of the assessments of the realism of the simulated scenario given by students and teachers. Table 4 shows the main descriptive values of both samples. The Kruskal–Wallis test checks whether there are significant differences between students and teachers, obtaining significant differences in all cases.

Table 5 shows confidence intervals for the difference between the mean rating of students and teachers, as well as the individual estimates of these differences. As can be seen, all the ratings are around 2 points higher for students than for teachers.

Figure 4 shows the differences detected numerically between students and teachers. In general, students evaluate all the variables considered with higher grades and there is a greater dispersion in their values.

Figure 5 shows the densities of each of the perceptions for students and teachers. In this graph it can be seen that the values assigned by the students tend to be higher than the teachers’ evaluations (as had already been concluded from the Kruskal–cWallis test). The perceived realism of the manikin has a uniform valuation by the students, while for the teachers it has a valuation with very little standard deviation.

### 3.4. Connection between Students’ Perception of Competence and Realisms Perceived by Students and Teachers

Specifically, Table 6 shows the correlations between the realism measured by the samples of students and teachers, where it was found that there is a significant correlation between the perception of clinical competence and the global realism perceived by the students, this correlation being positive. This means that, as perceived global realism increases, the perception of subsequent clinical competence also increases.

However, when analyzing the existence of correlation between the global realism achieved (analyzed by the teachers) and the perception of clinical competencies by the students, the Spearman correlation and the *p*-values associated with the correlation test showed *p*-values greater than 0.05, so there is no significant relationship between global realism and the perception of clinical competencies. 

## 4. Discussion

Simulation is a fundamental teaching strategy in health sciences education because it can incorporate technological advances by reproducing clinical scenarios in a safe environment, both for the patient and the student. It is considered an excellent teaching method, even though it also presents some difficulties such as a high workload for scenario development, the complexity of achieving an adequate correlation between scenario objectives and program competencies, the time dedicated and the student/teacher ratio [2]. 

In order to achieve efficient learning, it is necessary for the student to perceive the scenario as real, achieving immersion in all the areas of knowledge that will lead to the most accurate decision making. Several factors can increase this immersion. Make-up is one of the techniques that add realism to the scenarios and should therefore be included in medical education [18], since its authenticity contributes to greater commitment, highlighting the importance of the activity [19]. Contextualization (knowing the social and cultural circumstances on which one is acting) is another factor which increases environmental fidelity favoring the experience [9] and improving both the students’ immersion and clinical performance [20].

The degree of realism can be measured through different scales. Grahan and McAlee point out how the outcome of educational interventions should assess 4 levels: reaction, learning, behavior, and results. They also show that through a realistic evaluation all aspects of the intervention can be explored, including expected actions, unexpected side effects, positive actions that increase knowledge and negative ones [21]. Hagiwara et al. [22] created the “Immersion Score Rating Instrument (ISRI)”, which is a scale that analyzes 10 fundamental events or signs referring to: the instructor’s intervention, problems with the equipment, interaction with the manikins, technological distracters, responses to stimuli... so that 7 indicate a reduction in the student’s immersion and 3 favor it [22]. In the present study, we used the validated ProRealSim v.1.0 tool for the testing of the simulator by two teachers. This tool measures the fidelity of all the components of the scenario [23].

Some authors believe that there should be a high degree of fidelity to promote learner immersion; however, others believe that a high level of realism may overestimate the student’s abilities [3], so we should admit that all degrees of fidelity and/or realism can be beneficial if used appropriately [3]. Despite the importance of fidelity in scenario building, some authors do not consider it essential for achieving learning outcomes [5].

In addition, the level of realism is not always perceived in the same way by all participants, nor is the value given the same for each of them. Thus, a study carried out with scenarios involving virtual patients showed that some participants considered it to be realistic and others did not and concluded that greater realism is needed to increase the commitment of all participants [24].

Usually, teachers tend to be more critical and demanding in their perceptions since they consider that, on many occasions, the manikins do not exactly replicate the physiological conditions of a real patient [16]. A study conducted by Abu Dabrh et al. [25] obtained similar perceptions between teachers and students; however, the perceptions of standardized patients were more positive than those of teachers, indicating that the latter result could be due to the fact that patients (actors) emphasize personal interactions while educators focus primarily on clinical knowledge. Our students perceived greater realism than the teachers overall, as well as in all the partial realism dimensions analyzed (simulated participant, dental office and simulator), this measure being about two points higher than the teachers’ evaluations. After the team’s experience of more than three years measuring realism with different theoretical and mathematical tools, it is considered and applied that the gold standard in the assessment of this variable should be provided by the team of teaching evaluators.

Based on these studies and the previous results of this research team, and although for this specific sample of students the manikin was realistic, preference was given to the teacher’s criterion, due to its greater scientific soundness, and it was decided to propose an improvement in the level of realism of the manikin. Following the trend of this and other studies, this would improve the student’s perception of realism with the formative level of this study and could satisfy the realism needs of more demanding students of other learning grades and with more experience.

Carrero-Planells et al. evaluated student and teacher satisfaction in a high-fidelity simulation. The teachers rated the experience as very rewarding for the student although they observed a high level of stress regardless of the difficulty of the scenario. Students perceived a high degree of realism (9.2 points out of 10) and rated the quality of the simulator very highly, while the degree of credibility during the performance was the lowest rated factor [26]. A perception of satisfaction with the fidelity of the simulation is also reported with models of medium fidelity, especially in the case of novice students [7].

In the present study, we found that the greatest realism perceived by both students and teachers was for the simulated participant (actress), although this result was not statistically significant in the sample of students and strongly significant in the sample of teachers. Similar results were reported by Meerdink and Khan, who carried out a study comparing scenarios performed with actors or with manikins and found that the use of actors was much more realistic, favoring learning results, especially in the aspect of communication [14].

These data can also be applied to virtual patients and are highly valued by both teachers and students to improve knowledge acquisition and clinical decision making [27]. Perhaps the problem for the credibility of the manikins is the type of learning that one wants to obtain. For this reason, some authors consider that the function of the simulator is more important than its anatomical appearance [28], and that the commitment of the student predominates over the likeness of the simulator, so that fidelity should be considered as a mere mediator for immersion and learning [22]. Accordingly, in our work, the manikin was the one with the worst perception of realism obtained by the students (6.79).

Regarding the competencies acquired, most studies consider that the participants not only increase their level of knowledge and self-confidence, but also increase their human skills [7,10,17,24,27,29,30,31]. They also consider it as an emotionally pleasurable experience and that it should therefore be included in medical programs [30]. In the study by Pirany et al. [29] an increase in communication skills, in the confidence to treat patients and in the management of rare cases and with uncooperative patients was observed. Hanshaw and Dickerson note how high fidelity increases critical thinking knowledge, self-efficacy, self-confidence, critical judgment, and motivation [32]. Roze et al. find how teachers noted an increase in motivation and long-term concept memorization in their students [30]. Carrero-Planells et al. in a survey of teachers and students after a simulation obtained a very high satisfaction index for both groups, related to the fidelity of the scenario [26]. Our study also found that as the student’s perception of global realism increased, the perception of clinical competence increased; however, realism as measured by the faculty was not significantly related to the student’s perception of clinical competence.

## 5. Limitations of the Study

Among the limitations of the study, we should point out fundamentally those referring to the simulator. In dentistry, at present, we do not have pediatric manikins that fulfill the adequate functionality to simulate the voice, emotional, gestural and verbal expressions, which should be substituted by the teacher or assistant, which could reduce the perception of realism by the student. Another limitation could be the lack of studies of this nature in pediatric dentistry teaching, which prevents the availability of a high volume of references provided by other authors to contrast the results obtained; hence the need to carry out similar studies with larger samples.

## 6. Conclusions

This study demonstrates that increasing realism in the creation of scenarios with a handmade simulator and professional actress increases the perception of students in the acquisition of clinical skills, finding significant differences between the level of competence acquired and the perceived realism. In both samples (teachers and students), the best realism was for the simulated patient and the worst for the manikin, with the perception of the latter better valued by the students than by the teacher. 

Despite the high levels of fidelity achieved through technological advances, further research is needed to determine the role of realism in student learning.

## Figures and Tables

**Figure 1 ijerph-19-11387-f001:**
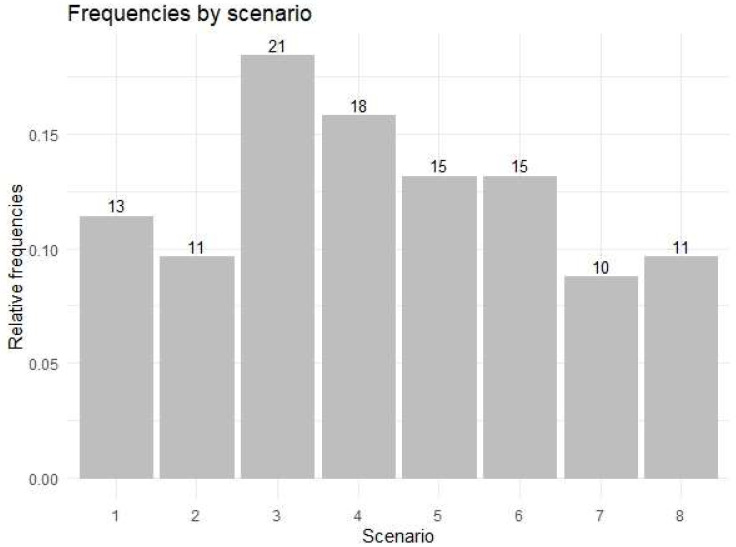
Bar graph of the different scenarios on the same case of pulp abscess in 8.5.

**Figure 2 ijerph-19-11387-f002:**
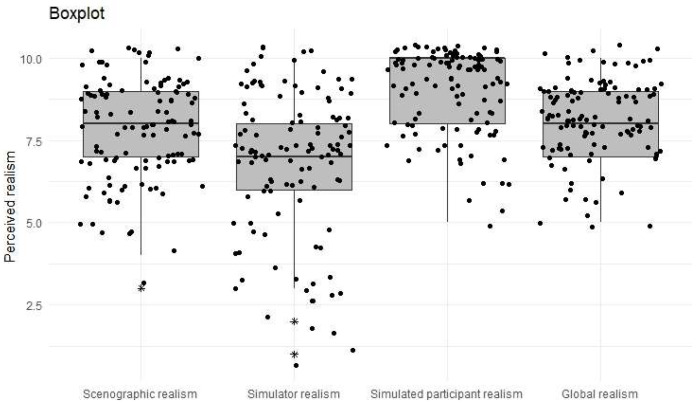
Box plot of realism perceived by students. Tests used Kruskal–Wallis test, ANOVA test for repeated means = 1.24 × 10^−^^2^^3^.

**Figure 3 ijerph-19-11387-f003:**
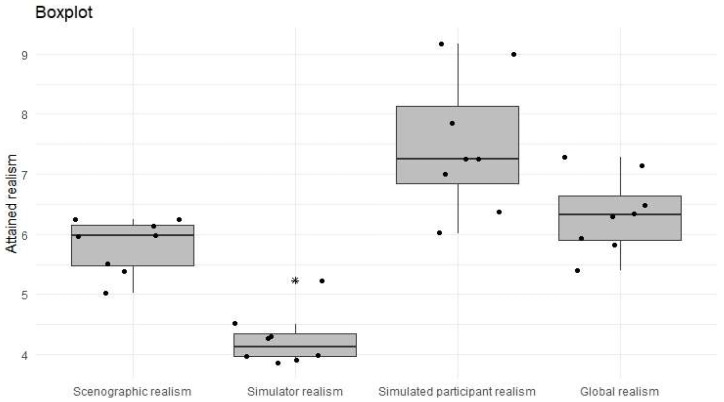
Box plot of realism as perceived by teachers.

**Figure 4 ijerph-19-11387-f004:**
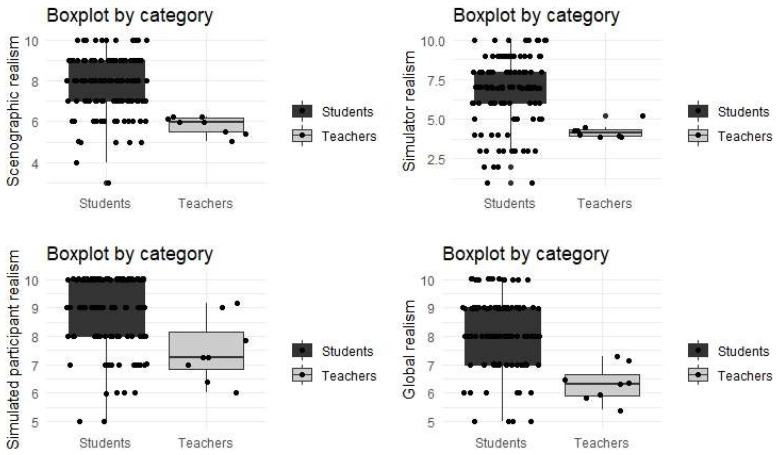
Box plot of realism as perceived by students and teachers.

**Figure 5 ijerph-19-11387-f005:**
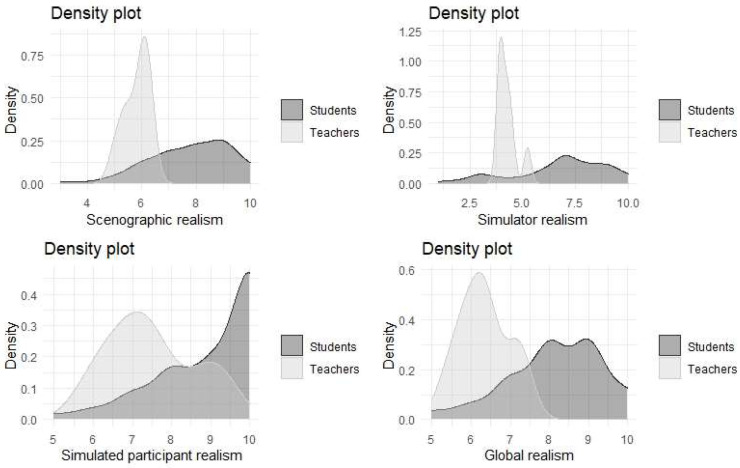
Densities of realism as perceived by students and teachers.

**Table 1 ijerph-19-11387-t001:** Questionnaire applied for students’ evaluation of scenario realism.

Perception of Realism (Postscenario)
How realistic did you find the dental office?
How realistic did you find the girl patient manikin?
How realistic did the mother seem to you?
How realistic did everything seem to you, overall?
**Perception of clinical competence (Posttest)**
How capable do you feel, at this point, to perform behavioral management of a pediatric patient and their parents, and to diagnose and correctly treat a pulp therapy case in a real pediatric dentistry practice?

**Table 2 ijerph-19-11387-t002:** Realism perceived by students in global terms.

	Perceived Realism of the Simulated Office			Perceived Realism of the Mannequin			Perceived Realism of the Simulated Participant			Perceived Global Realism		
Total Number of Scenarios	Media	Standard Deviation	*p*-Value	Mean	Standard Deviation	*p*-Value	Mean	Standard Deviation	*p*-Value	Mean	Standard Deviation	*p*-Value
8	7.833	1.48	0.009502 (a)	6.79	2.22	0.001456	8.99	1.28	0.7292 (a)	8.15	1.22	0.009441 (a)

Tests used (a) Kruskal Wallis test, ANOVA test for repeated means = 1.24 × 10^−^^2^^3^.

**Table 3 ijerph-19-11387-t003:** Realism perceived by teachers.

	Mean	Standard Deviation	*p*-Value
Perceived realism of the office	5.81	0.43	1.54 × 10^−32^ (a)
Perceived realism of the mannequin	4.25	0.45	1.54 × 10^−32^ (a)
Perceived realism of the participant	7.49	1.13	1.54 × 10^−32^ (a)
Global perceived realism	6.34	0.64	1.54 × 10^−32^ (a)

Test used (a) Friedman test.

**Table 4 ijerph-19-11387-t004:** Realism as perceived by students and teachers.

	Perceived Realism of the Office			Perceived Realism of the Mannequin			Perceived Realism of the Participant			Global Perceived Realism		
Position	Mean	Standard Deviation	*p*-Value	Mean	Standard Deviation	*p*-Value	Mean	Standard Deviation	*p*-Value	Mean	Standard Deviation	*p*-Value
Student	7.83	1.48	0.0001797 (a)	6.79	2.22	0.001181 (a)	8.99	1.28	0.001857 (a)	8.15	1.22	0.0001889 (a)
Teacher	5.81	0.453	0.0001797 (a)	4.25	0.454	0.001181 (a)	7.49	1.13	0.001857 (a)	6.34	0.641	0.0001889 (a)

Test used (a) Kruskal–Wallis.

**Table 5 ijerph-19-11387-t005:** Confidence intervals and individual estimates of the difference in mean student and faculty ratings.

	95% CI for Student-Faculty Variance Difference			
	Perceived Realism of the Office	Perceived Realism of the Mannequin	Perceived Realism of the Participant	Global Perceived Realism
Punctuation	2.020833	2.535724	1.499978	1.810373
95% IC	[1.579782, 2.461884]	[2.005260, 3.066188]	[0.5415388, 2.4584174]	[1.251806, 2.368940]

**Table 6 ijerph-19-11387-t006:** Correlation between realism as measured by students and teachers.

	Realism Measured by Students		Realism Measured by the Faculty	
	Spearman’s Correlation	*p*-Value	Spearman’s Correlation	*p*-Value
Student’s perception of clinical competence	0.26	0.00576 (a)	−0.1571209	0.09502 (a)

Test used (a) Spearman’s Correlation Test.

## Data Availability

Not applicable.

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
