# Peer review of "Perception of Realism and Acquisition of Clinical Skills in Simulated Pediatric Dentistry Scenarios"

_ijerph, 2022, doi:10.3390/ijerph191811387_

Round 1

Reviewer 1 Report

Overall the paper presents an interesting topic with a need for further development.

Some lacking areas of the manuscript are that they only used 2 teachers, and there is no mention of the calibration process for them. Additionally, this study is based on subjective perceptions and lacks scientific evidence.

Author Response

The research team is deeply grateful for the reviewer's contributions and has made all the adjustments indicated by him in the article.

Explanation of the calibration process of the validated tool and justification of use of the two evaluators: Lines 148 to 155

Any evaluation is a subjective expression, but when it is linked to the indicated statistical studies and validated indexes, as in this article, the results attain scientific validity and reliability. This explanation has been added in lines 157-163.

The reviewer's contributions have greatly improved the article. We are very grateful.

Having made the above improvements to the body of the manuscript, we await a further revision.

Reviewer 2 Report

Esteemed colleagues I congratulate you on the effort of improving student educations through simulation.

However, there are some aspects regarding your manuscript that require your attention:

-        You mention 8 scenarios, I believe it will be better to describe each scenario, the level of difficulty of the procedure that the student had to perform on the mannequin. This is very important to clearly visualize the setting and the possible design flaws of the simulation scenario.

-        Also why did you take a sample of 114 students, why not present the data of all 283 students?

-        Again, if you chose a sample of students why not sample an equal number of individuals for each scenario and therefore you could probably compare the data obtained between different scenarios?

-        Also, you compare the data of the 2 examiners with the data from the students, this seems an approach with little statistical support – you have 2 teachers and 114 students. Moreover, FI cand draw the conclusion that the students are satisfied with the simulator and the only reason for improving the simulator is only the dissatisfaction of the teachers?

-        Figures 5 and 6 could be spread into multiple figures in order to better visualize the data, I had to magnify the PDF at 300% in order to clearly view the results.

I believe that after solving these problems the manuscript will be greatly improved and I am looking forward to review the next version of the manuscript.

Reviewer 3 Report

Dear Authors,

This is a very important and interesting study. The purpose of the questions involved in the original research cover significant issues and attract many readers in the fields of pediatric dentistry. However, the study has limitations in both the introduction and conclusion sections where they need further improvements. Plus, the study involves lots of grammatical mistakes where they require correction by experts. The below points are suggestions to be considered for improving the quality of the research in the study and attracting valuable readers. 

·       The abstract: remove space L19. How you calculate the number of participants? Conclusion of the abstract needs reformulation you should be objective not subjective, in addition, how the reader can know from the abstract the increased number of 0.956 how you calculate this number. Lack of objective explanation in abstracts. Where is the P-value, sample size calculation?

·       The authors should add and organize the keywords according to mesh terms and alphabetical numbers.

·       The introduction is adequately written, but with insufficient information about the topic. L35 remove these … L41: Please add references. Remove space L46. The paragraph talking about fidelity should be reformulated and not all ideas should be taken from reference number 3. Please remove “we” or “our” in the manuscript

·       Materials and methods well written. Please put the full name of ANOVA before abbreviation.

·       Figure 2 should be replaced by another high-resolution image.

·       For statistical significance letters should be added to tables.

·       Please include more limitations on the study. 

·       Reformulate the conclusion section.

·       References need further checking. 

·       I am not a native English speaker, but I could easily notice a lot of English mistakes in the manuscript. English revision is a must.

Round 2

Reviewer 2 Report

You answered all the observations of the reviewers.

Reviewer 3 Report

Thank you for your answers. The paper is now suitable for publication.